# Micro-RNAs from Plasma-Derived Small Extracellular Vesicles as Potential Biomarkers for Tic Disorders Diagnosis

**DOI:** 10.3390/brainsci12070829

**Published:** 2022-06-26

**Authors:** Yilong Wang, Xuebin Xu, Haihua Chen, Mengying Zhu, Xiaotong Guo, Feng Gao

**Affiliations:** 1Department of Neurology, Children’s Hospital, Zhejiang University School of Medicine, Hangzhou 310052, China; yilongwang@zju.edu.cn (Y.W.); 22118472@zju.edu.cn (X.X.); chh0201323@163.com (H.C.); 0621725@zju.edu.cn (M.Z.); 22118484@zju.edu.cn (X.G.); 2National Clinical Research Center for Child Health, Children’s Hospital, Zhejiang University School of Medicine, Hangzhou 310052, China

**Keywords:** tic disorder, Tourette Syndrome, chronic motor or vocal tic disorder, transient tic disorder, small extracellular vesicles, miRNAs, diagnosis marker

## Abstract

Tic disorders (TDs) are a series of childhood neuropsychiatric disorders characterized by involuntary motor and/or vocal tics and commonly comorbid with several other psychopathological and/or behavioral disorders (e.g., attention deficit hyperactivity disorder and obsessive–compulsive disorder), which indeed aggravate clinical symptoms and complicate diagnosis and treatment. Micro-RNAs (miRNAs) derived from small extracellular vesicles (sEVs) have been recognized as novel circulating biomarkers of disease. To identify specific miRNAs derived from plasma sEVs for TDs’ diagnosis and prognosis, we used official EV isolation and purification methods to characterize the plasma-derived EV miRNAs from children with different types of TDs. Nanoparticle tracking analysis, transmission electron microscopy, and immunoblot analysis of EV surface markers were applied to confirm the features and quality of sEVs. The RNA sequencing (RNA-seq) approach was adapted to identify novel circulating sEVs-derived miRNAs with altered expression levels in paired comparisons of TDs versus healthy controls (HCs), transient tic disorder (TTD) versus chronic motor or vocal tic disorder (CTD), and TTD versus Tourette Syndrome (TS). GO term and KEGG pathway were performed for functional analysis and the receiver operator curve analysis was followed to test the diagnosis efficacy of differentially expressed miRNAs (DEMs) derived from plasma sEVs among paired groups, namely, TDs versus HCs, TTD versus CTD, and TTD versus TS. As a result, 10 miRNAs (hsa-let-7a, hsa-let-7b, hsa-let-7c, hsa-let-7e, hsa-let-7f, hsa-miR-25-3p, hsa-miR-29a-3p, hsa-miR-30b-5p, hsa-miR-125b-5p, and hsa-miR-1469) have demonstrated a significantly different expression signature in the TDs group compared to HCs with excellent area under curve (AUC) values of 0.99, 0.973, 0.997, 1, 0.99, 0.997, 0.987, 0.993, 0.977, and 0.997, respectively, and the diagnostic efficacy of miRNAs was also estimated for discriminating TTD from CTD or TS. In our research, we finally obtained several potential sEVs-derived miRNA biomarkers to assess the diagnosis and prognosis of TDs.

## 1. Introduction

Tic disorders (TDs) are a class of childhood–adolescence-onset neuropsychiatric disorders defined by involuntary motor and or vocal tics. The etiology of TDs is recognized as the interaction of genetic, immunological, psychological, and environmental factors [1]. TDs are classified into transient tic disorder (TTD), chronic motor or vocal tic disorder (CTD), Tourette Syndrome (TS), and tic disorder not otherwise specified (NOS), in terms of the clinical manifestations (motor or vocal, or a combination of both) and duration of the disease according to the fifth revision of the Diagnostic and Statistical Manual of Mental Disorders (DSM-5) [2]. TDs are usually comorbid with several other psychopathological and/or behavioral disorders, including attention deficit hyperactivity disorder (ADHD), obsessive–compulsive behavior (OCB) or disorder (OCD), and so forth [3]. According to the results of a national-scale psychiatric epidemiological survey, the overall prevalence of TDs was 2.5% (95% CI: 2.3–3.0) in 173,992 participants aged 6–16 years from 58 rural schools and 113 urban schools in China, and the children 6–11 years of age had a significantly higher prevalence than the younger group of adolescents aged 12–16 years [4]. The involuntary tics and comorbid psychopathological and behavioral conditions will lead to remarkable damage in family and social interactions and school and job performance, and thus adversely impair patients and their family’s quality of life [5]. Tic severity, premonitory urges, and a family history of TDs in childhood were identified as predictors of a poorer health-related quality of life (HR-QoL) in adults [6]. HR-QoL profiles in children with TDs are often impaired by comorbid attention-deficit and hyperactivity symptoms, which tend to improve with age, and adults’ perception of QoL seems to reflect a greater impact from depression and anxiety symptoms [7]. Therefore, early childhood evaluation, intervention, and therapy have been indicated to improve symptoms and reduce complications of TDs in adulthood, guaranteeing an overall superior quality of life [5]. The various manifestations of TDs and the syndromes of comorbid conditions complicate the diagnosis of TDs [8]. The lack of objective diagnostic tests with a strong dependence on a behavioral diagnosis and psychometric evaluations [9,10,11] increases the difficulty for clinicians to complete an early diagnosis of TDs. Recently, differential expression of micro-RNAs (miRNAs) as molecular evidence demonstrated the potential in the accurate and objective diagnosis of neurological and psychopathological disorders [12].

MiRNAs are small, conserved, non-protein-encoding RNA molecules (18–28 nucleotides in length) that are responsible for the regulation of protein expression on post-transcriptional levels [13]. Individual miRNAs have been reported to regulate multiple biological functions including cell differentiation, developmental timings, embryogenesis, metabolism, and apoptosis [14]. MiRNAs have currently been confirmed to facilitate diagnosis as biomarkers that can discriminate illness from health. Numerous studies have indicated differential miRNA expression profiles in multiple cancer types, such as leukemia, breast cancer, and ovarian cancer [15,16,17]. In neurological disorders, nearly 70% of the miRNAs are expressed in the human cerebrum and many of them are particular to neurons [18]. Distinctive miRNA expression is involved in neuronal differentiation and morphological properties [19] and contributes to ir-regular cerebrum development and the mechanism of multiple neurodevelopmental disorders [20], such as Alzheimer’s disease [21], Parkinson’s disease [22], epilepsy [23,24], and Tourette Syndrome [25]. The assessment of miRNA expression level in patients with nervous system disorders may help diagnose related neurological disease. In recent years, altered plasma-derived miRNAs profiles have been detected in TS children through a high throughput technique, and miRNA-189 [25] and miRNA-429 [8] have been reported as indicators in diagnosis TS. Despite several studies conducted on the correlation between the plasma miRNAs and diagnosis of TS, there is still a lack of work on identifying specific miRNAs as biomarkers for the diagnosis of other types of TD, and research on biomarkers for TS is also extremely scarce [12].

Both prokaryotes and eukaryote extrude extracellular vesicles (EVs) are membrane-bound organelles encompassing various types of molecular contents (e.g., nucleic acids, protein, and metabolites) [26]. EVs are supposed to function in intercellular interactions by transferring their contents (including RNA, DNA, lipids, proteins, etc.) to targeted cells, such as immune reactions, viral pathogenicity, neurological disorders, and oncology [27]. Nevertheless, the cargos are also protected in encapsulation by EVs [28,29]. The Western blotting and flow cytometry assay of EVs surface markers, such as CD 63, Tsg101, and HSP 70, should be performed for the identification of EVs [30]. The classic dichotomy of extracellular vesicles has relied on size and biogenesis and small EVs (sEVs) range from ~40 to 160 nm in size. Considering the importance of sEVs and sEVs-derived miRNAs in intercellular signaling and cell-to-cell communication, there is increasing interest in their potential role as noninvasive biomarkers for disease detection and prognosis. sEVs-derived miRNAs in human bodily fluids have been reported to demonstrate significantly different expression profiles between patients with specific diseases (cancer, cardiometabolic disease, infection disease, neurologic disease, etc.) and healthy controls (HCs) [27]. Unfortunately, a dysregulated miRNAs panel has not yet been established as a tool for TD diagnostics. 

We assumed that altered sEVs-derived miRNAs profiles in plasma could be detected from patients with TD using an array of bioinformatics instruments. This molecular evidence could be directly applicable for enhancing traditional TD diagnosis when combined with DSM-V criteria, and equally important, it will also afford potential therapeutic targets for TD. This study aims to provide more accurate and faithful biomarkers that can assist in the diagnosis and therapy of TD. 

## 2. Materials and Methods

### 2.1. Participants’ Clinical Characteristics

The setting for this study is Children’s Hospital of Zhejiang University School of Medicine, and ethics approval for this study was granted by our hospital Ethics Committee (2022-IRB-027). The parents or guardians of all children in this study signed written informed consent before data were collected.

A total of 30 pediatric participants who were clinically diagnosed with tic disorders (10 with Tourette Syndrome, 10 with transient tic disorder, and 10 with chronic motor or vocal tic disorder) at the outpatient facility between April 2021 and December 2021 were enrolled in this study along with 10 age- and gender-matched children as a healthy control group. The healthy control subjects were recruited from the Children’s Hospital of Zhejiang University School of Medicine, and confirmed healthy and neurologically typical by medical history, general examinations, and laboratory examinations, and all had been excluded from ADHD and OCD screening. All blood samples of these participants were collected by experienced and skilled nurses. Plasma miRNAs from sEVs between tic disorder patients and healthy controls were sequenced at first using Illumina NovaSeq 6000 technology in this study. All tic disorder participants were diagnosed by the Department of Neurology in our hospital. All patients met the inclusion criteria based on DSM-5 as follows: (a) People with Tourette Syndrome have both motor and vocal tics, and have had tic symptoms for at least 1 year. (b) People with chronic motor or vocal tic disorders have either motor or vocal tics, and have had tic symptoms for at least 1 year. (c) People with transient tic disorder can have motor or vocal tics, or both, but have had their symptoms less than 1 year, and have no recurrence after a year.

### 2.2. Blood Collection and Isolation of Small Extracellular Vesicles (sEV)

In this study, peripheral venous blood specimen from Tourette Syndrome patients, transient tic disorder children, chronic motor or vocal tic disorder children, and HCs were randomly collected and processed. Combination size-exclusion chromatography (SEC) with ultrafiltration was considered a kind of low-recovery and high-specificity method for the isolation of sEVs [29]. All blood samples were collected in EDTA tubes, centrifuged at 3000× *g* for 15 min at 4 °C to separate the plasma, and stored at −80 °C for subsequent experiments. The sEVs isolation was performed using SEC with minor modifications. In short, 1 mL of plasma sample filtered through a 0.8-μm microporous membrane filter was transferred to an Exosupur^®^ column (Echobiotech, Beijing, China). Elusion was performed using PBS, and 2 mL of eluate was collected following the manufacturer’s instructions. The fractions were collected by filtering through a 100-kD molecular weight cutoff spin column of Amicon^®^ Ultra spin filters (Merck, Germany) to a final volume of 200 µL. 

### 2.3. Nanoparticle Tracking Analysis (NTA)

Vesicles were evaluated in suspension at concentrations ranging from 1 × 10^7^/mL to 1 × 10^9^/mL using a ZetaView PMX 110 (Particle Metrix, Meerbusch, Germany) equipped with a 405 nm laser source to confirm the size and number of isolated vesicles. Then, 60 s of video at 30 frames per second were recorded and vesicle movements were tracked using NTA software ZetaView version 8.02.28 (Particle Metrix, Meerbusch, Germany).

### 2.4. Transmission Electron Microscopy (TEM)

Using TEM, we characterized the morphology of sEVs [31]. A total of 10 µL of plasma sEV samples were placed on copper grids and incubated at 20 °C for 1 min. Plasma sEV samples were then treated with uranyl acetate solution for 1 min and washed with sterile, distilled water. Water was removed from the sample and the sample was kept under incandescent light for 2 min. The copper mesh was monitored and images were taken using a transmission electron microscope (H-7650, Hitachi Ltd., Tokyo, Japan).

### 2.5. Western Blot Assay

Western blot analysis for identification of the presence of the sEVs was performed following the established procedures [32]. sEV supernatants were denatured in 5× sodium dodecyl sulfate (SDS) buffer for 5 min at 95 °C prior to electrophoresis in 10% SDS-polyacrylamide gel using 50 µg of total protein per lane. Antibodies CD63 (Santa, sc-5275), HSP 70 (Abcam, ab181606), TSG101 (Abcam, ab125011), and calnexin (Proteintech, 10427-2) were used to detect specific proteins. Among these markers, calnexin (Proteintech, 10427-2) was the marker present only in the endoplasmic reticulum but not in EVs, and antibodies CD63 (Santa, sc-5275), HSP 70 (Abcam, ab181606), and TSG101 (Abcam, ab125011) were markers present only in EVs but not in other subcellular organelles. Cell lysates derived from mesenchymal stem cell (MSC) were used as control.

### 2.6. Library Preparation and Sequencing

Library preparation and sequencing were conducted for detecting the expression of sEVs-derived miRNAs [33]. Total RNA samples were extracted and purified from sEVs by the miRNeasy^®^ Mini kit (Qiagen, cat. No. 217004) according to the manufacturer’s instructions. The concentration and purity were assessed using the RNA Nano 6000 Assay Kit of Agilent Bioanalyzer 2100 System (Agilent Technologies, CA, USA). For small RNA libraries, a total of 1–500 ng of RNA per sample were prepared as input material. Sequencing libraries were constructed using QIAseq miRNA Library Kit (Qiagen, Frederick, MD, USA) in accordance with the manufacturer’s instructions. Reverse transcription (RT) primers with unique molecular indices (UMI) were introduced to analyze the quantification of miRNA expression during cDNA synthesis and PCR amplification. The Agilent Bioanalyzer 2100 and qPCR technique were used to evaluate the library quality. Cluster analysis of index-encoded products was performed using the acBot Cluster Generation System and TruSeq PE Cluster Kitv3-cBot-HS (Illumina, San Diego, CA, USA) following the manufacturer’s protocol. After cluster production, library preparations were analyzed using the Illumina NovaSeq 6000 system and paired-end reads were generated.

### 2.7. Bioinformatics Analysis of miRNA 

Expression profiles of sEVs-derived miRNAs were identified using the following bioinformation analysis. the raw data of miRNA sequencing were firstly processed through in-house perl scripts. In this step, clean reads were obtained by removing reads containing adapter, reads con-taining ploy-N and low-quality reads from raw data. And reads were trimmed and cleaned by removing the sequences smaller than 15 nt or longer than 35 nt. At the same time, Q20, Q30, GC-content and sequence duplication level of the clean data were cal-culated. All the downstream analyses were based on clean data with high quality.The clean reads with high quality were aligned with Silva database, GtRNAdb database, Rfam database, and Repbase database using Bowtie software, respectively, to refine rRNA, tRNA, small nuclear RNA (snRNA), small nucleolar RNA (snoRNA), other noncoding RNAs, and repeat sequences. The remaining reads were employed to detect known miRNAs and novel miRNAs predicted in comparison to known miRNAs in miRbase and Human Genome (GRCh38). The miRNAs expressions were corrected according to UMI counts and normalized to transcript per million (TPM). 

### 2.8. Gene Ontology (GO) Annotation and Kyoto Encyclopedia of Genes and Genomes (KEGG) Pathway Enrichment Analysis

The functional analysis of significantly dysregulated miRNA expression was realized using GO terms and KEGG pathway enrichment analysis. Prediction of the interactions between differentially expressed miRNAs (DEMs) and their targets was realized using miRanda (v3.3) (Anaconda, New York, NY, USA) and RNAhybrid (v2.1.1) (Bielefeld Bioinformatics Server, Bielefeld University, Bielefeld, Germany). The target genes of DEMs were subjected to GO enrichment analysis for functional annotation using the top GO R packages. The statistical enrichment analysis for target genes of DMEs was performed in KEGG pathways [34] (http://www.genome.jp/kegg/ accessed on 20 December 2020) was implemented by the KOBAS [35] software.

### 2.9. Statistical Analysis

Student’s *t*-tests were used to compare the results of processed clinical data between groups using SPSS 28.0.1 software (IBM, Armonk, NY, USA). Statistical tests to compare the miRNA expression levels and imaging features were conducted using R 3.5.1 (www.r-project.org accessed on 2 July 2018). The *p* value was determined using Compare Means in a ggpubr package with the *t*-test method to compare means. The receiver operator characteristic (ROC) curve was plotted using the procedures in pROC package. The area under curve (AUC) values were then estimated for comparing diagnostic efficiency of different sEV-miRNAs. *p* < 0.05 was considered as significant (not significant (NS), *p* > 0.05).

## 3. Results

### 3.1. Clinical Features of Participants

In total, 40 children were enrolled in the study, of whom 30 had tic disorders (8.36 ± 2.40), sex M: F = 25:5) and 10 were healthy controls (HCs) (mean age 9.49 ± 2.8), sex M: F = 7:3) (Table 1). There were more male than female participants in this study and the gender ratios between groups are very close to each other. Generally, boys are more frequently diagnosed with TD and its subtypes than girls [36]. In this study, the children endured several typical TD symptoms; 10 children (33.33%) had involuntary blinking, 4 (13.33%) involuntary grimacing, 8 (26.67%) involuntary throat clearing/phonation, 4 (13.33%) involuntary shoulders shrugging, 2 (6.67%) involuntary head shaking, and 2 (6.67%) involuntary spitting. In accordance with DSM-V TD criteria, 10 children had a diagnosis of TS, 10 CTD, and 10 TTD (Figure 1). Additionally, HCs recruited in this study had no personal or family history of TDs.

### 3.2. Isolation and Characterization of Human Plasma-Derived sEVs 

We performed initial experiments for isolation and characterization of sEVs from human plasma. Pretreated samples were loaded on the size-exclusion chromatography (SEC) for the separation of sEV-enriched fractions. The particle size and morphology of the vesicles were obtained using transmission electron microscopy (TEM) (Figure 2A), and the results of nanoparticle tracking analysis (NTA) indicates that the range of sEVs particle size is 75–200 nm (Figure 2B). Several proteins that were previously described as typical constituents of sEVs, such as CD63, TSG101, and HSP70, were identified in the isolated vesicles using Western blot analysis, while negative marker calnexin was not identified in fractions (Figure 2C). Thus, the purified sEVs were prepared for the following experiments. 

### 3.3. Identification of Differentially Expressed miRNAs in Plasma sEVs from the Tic Disorder Group and HCs 

We first tested 40 plasma samples (30 TDs and 10 controls) for characterization of the sEVs-derived miRNA from the plasma of the participants in the testing set using miRNA sequencing. In general, each sample generated 20 million reads, and a total of 1741 known miRNAs were detected from the sequencing data. The miRNAs’ quality was assured because miRNAs with low expression (TPM values less than ten) were deleted. Condition screening: *p*-adjust < 0.05 and |log2(fold change)| > 0.584. sEVs-derived miRNAs’ abundance is relatively low in human plasma samples [37,38]; thus, a 1.5-fold change (FC) in miRNAs’ expression level could be a cutoff for establishing differential change in this study. In the TD group, 126 DEMs were detected in plasma sEVs compared to HCs, with 65 upregulated genes and 61 downregulated genes (Figure 3A,B). In the TTD group, 96 miRNAs were differentially expressed in plasma sEVs compared to HCs with 63 miRNAs upregulated and 32 miRNAs downregulated. In the CTD group, 112 DEMs were detected in plasma sEVs compared to HCs with 69 upregulated genes and 43 downregulated genes. In the TS group, a total of 101 miRNAs were differentially expressed in plasma sEVs compared to HCs, with 39 miRNAs upregulated and 62 miRNAs downregulated. Consequently, a total of 205 DEMs were obtained from sEVs-derived miRNAs among four paired comparisons, namely, HC versus TDs, HC versus TS, HC versus TTD, and HC versus CTD (Figure 3C). Venn-diagrams demonstrated an overlapping pattern of miRNA sets with 36 upregulated or downregulated miRNA among these four groups (Figure 3C), while 18 miRNAs were commonly upregulated and 18 miRNAs were commonly downregulated, as illustrated in Figure 3D using the bar chart. Subsequently, GO and KEGG pathway analyses were performed for determining the potential functions of these DEMs in HC versus TDs. The results demonstrated that 12,880 mRNAs targeted by 126 DEMs were enriched in the neuron differentiation, neurogenesis, nervous system development, generation of neuron, neuronal cell body, neuron projection, synapse Ras GTPase binding, Axon guidance, and toxoplasmosis (Figure 3(E1–E4)).

### 3.4. Diagnostic Potential for sEVs-Derived miRNAs by ROC Curve Analysis

We further performed receiver operating characteristic (ROC) curve analysis to verify the diagnosis efficacy of 36 overlapping miRNAs between the TD group and HCs. Then, the area under the curve (AUC) was calculated immediately. Five upregulated miRNAs (hsa-let-7a, hsa-let-7b, hsa-let-7c, hsa-let-7e, and hsa-let-7f) exhibited an AUC of 0.99, 0.973, 0.997, 1, and 0.99, respectively. Another five downregulated miRNAs (hsa-miR-25-3p, hsa-miR-29a-3p, hsa-miR-30b-5p, hsa-miR-125b-5p, and hsa-miR-1469) demonstrated an AUC of 0.997, 0.987, 0.993, 0.977, and 0.997, respectively (Figure 4). 

### 3.5. Identification of Differentially Expressed miRNAs in Plasma sEVs from the TTD, CTD, and TS Groups

We then analyzed the DEMs from the TTD group vs. the CTD group and the TTD group vs. the TS group to seek the prognosis markers. Additionally, 13 miRNAs were upregulated, and another 27 miRNAs were downregulated in the TTD group compared with the CTD group, and the levels of 9 miRNAs were increased, and those of 36 miRNAs were decreased in the TTD group in comparison to the TS group (Figure 5A,B). Furthermore, 17 common DEMs were detected in two paired comparisons, the TTD group versus the CTD group and the TTD group versus the TS group (Figure 5C). 

The diagnosis efficacy of 17 overlapping differentially expressed miRNAs among these two groups was also assessed with ROC curve analysis. As a result, upregulated hsa-miR-365b-3p achieved an AUC of 0.88 both in TTD versus TS and TTD versus CTD comparisons. Another miRNA, hsa-miR-485-5p, demonstrated an AUC of 0.9 in the TTD group compared to TS group with downregulated expression, and a higher value of AUC (0.79) in the TTD group compared to the CTD group. Downregulated hsa-miR-6882-5p also indicated a good value of AUC (0.8) in the TTD group compared to the TS group, and an AUC of 0.77 in the TTD group compared to the CTD group (Figure 5(C3)).

## 4. Discussion

In the nervous system, small extracellular vesicles (sEVs) generated by all cell types in the brain, including neurons [27,28], astrocytes [29,30], oligodendrocytes [31,32], as well as microglia [33], are considered to play a critical role in mediating intercellular communication, and the application of sEVs in the study of the central nervous system is gradually increasing [39]. sEVs-derived miRNAs have recently emerged as potential biomarkers for the rapid and accurate diagnosis of multiple CNS diseases [40]. It has been reported that sEVs-derived miRNAs are packaged into membrane-bound vesicles selectively in a specific process [41,42]. As a result, sEVs-derived miRNA’s profile was designated and specified previously. In addition, membrane-enclosed vesicles could effectively protect contained miRNAs from the digestion by RNase in biofluids [43] and thus, sEVs-derived miRNAs are more reliable than circulating cell-free miRNAs. In previous research, miRNAs such as hsa-miR-429 and hsa-miR-189 that indicated potential in TDs diagnosis had already been widely studied and further estimated [8,12]. Thus, we hypothesized that EVs-derived miRNAs may contribute to the diagnosis and prognosis of TDs. Here, we identified differentially expressed miRNAs from the plasma-derived sEVs in TDs versus HCs using RNA-seq and bioinformatics analysis.

In our study, plasma-derived sEVs from children were isolated using ultrafiltration combined with size-exclusion chromatography (SEC), which was recognized as a method with high specificity and low recovery. Additionally, 36 differentially expressed miRNAs (DEMs) were selected between TDs and controls with 18 upregulated miRNAs and 18 downregulated miRNAs. Furthermore, 10 of these DEMs (hsa-let-7a, hsa-let-7b, hsa-let-7c, hsa-let-7e, hsa-let-7f, hsa-miR-25-3p, hsa-miR-29a-3p, hsa-miR-30b-5p, hsa-miR-125b-5p, and hsa-miR-1469) were finally selected for receiver operating curve (ROC) analysis, which could distinguish TD patients from HCs with an excellent range of AUC values from 0.973 to 1.

The early detection and accurate diagnosis of TS and CTD will improve the prognosis of the patients. Thus, we subsequently seek the sEVs-derived miRNAs as prognosis markers from the TTD, the CTD, and the TS groups, and 17 common miRNAs with dysregulated expression were identified both in two comparisons, TTD versus TS, and TTD versus CTD. 3 DEMs (hsa-miR-365b-3p, hsa-miR-485-5p, and hsa-miR-6882-5p) were finally selected for ROC analysis and achieved good diagnosis efficacy in differentiating patients with TTDs from patients with TS or CTDs.

Previous studies showed that hsa-let-7a and hsa-let-7d acted as noninvasive diagnostic biomarkers for diabetic retinopathy and lung carcinoma, respectively [44,45]. In addition, hsa-let-7a and hsa-let-7b were reported to have significant diagnostic values for detecting atrial septal defects [46]. Research confirmed that hsa-miR-25-3p and hsa-miR-30b can be used as a potential biomarker for breast cancer [47]. Hsa-miR-125b-5p was stably upregulated in the plasma from patients with alveolar echinococcosis (AE), indicating a promising biomarker for early, noninvasive diagnosis of AE [48]. To date, our study is the first to validate the significant upregulation of hsa-let-7a, hsa-let-7b, hsa-let-7c, hsa-let-7e, and hsa-let-7f, and downregulation of hsa-miR-25-3p, hsa-miR-29a-3p, hsa-miR-30b-5p, hsa-miR-125b-5p, and hsa-miR-1469 in plasma samples from patients with TDs, suggesting the potential usefulness of these miRNAs as novel, noninvasive, and independent biomarkers for TDs diagnosis. Furthermore, we found that hsa-miR-365b-3p, hsa-miR-485-5p, and hsa-miR-6882-5p demonstrated promising diagnosis efficacy in discriminating TTD from TS or CTD. The DEMs identified in our study did not show any overlap with previous reports. Up to now, only two miRNAs, hsa-miR-189 and hsa-miR-429, had been reported in previous studies of TDs. Increased expression of hsa-miR-189 had been found by Abelson et al. in TS patients [25]. Renata et al. found that hsa-miR-429 was significantly downregulated in serum from TS patients [8]. These different findings may be explained by the following reasons: (i) the different selection standards of children with TS; (ii) different screening conditions for dysregulated miRNAs; (iii) different sample type and size; (iiii) individual characteristics including race, lifestyle, etc.

Recent decades of studies have reported a significant genetic contribution to most of TDs, especially TS. Several mRNAs and their corresponding miRNAs, which have been reported to play a critical role in TDs, were acquired and summarized by searching the database. According to the table, 12 miRNAs (hsa-miR-125a-3p, hsa-miR-1260b, hsa-miR-760, hsa-miR-939-5p, hsa-miR-6754-3p, hsa-miR-146b-3p, hsa-miR-339-5p, hsa-miR-6724-5p, hsa-miR-6724-5p, hsa-miR-939-5p, hsa-let-7a-5p, and hsa-miR-6738-5p) of 126 DEMs have been shown to target mRNA expression of DRD4, STX1A, SLC6A4, PNKD, and DRD2 genes, which might be related to TDs. Sun et al. found that PNKD factored majorly in neuronal development and function [49]. Hsa-miR-6738-5p might be involved in the transduction pathways of TDs. In another study, DRD2 mRNA expression level in peripheral blood lymphocytes have potential in the diagnosis and disease assessment of TDs [50]. Hsa-miR-6724-5p, hsa-miR-939-5p, and hsa-let-7a-5p have been found to target mRNA expression of DRD2. The findings in the current study indicated that multiple variants in SLC6A4 were responsible for TDs, characterized by obligatory/unwanted movements, behaviors, or thoughts, while mRNA expression of SLC6A4 might be targeted by hsa-miR-146b-3p, hsa-miR-339-5p, and hsa-miR-6724-5p [51]. In addition, some studies have shown that DRD4 is a genetic risk factor of severe symptoms in children with TDs [52]. The corresponding miRNAs are hsa-miR-760, hsa-miR-939-5p, hsa-miR-125a-3p, and hsa-miR-1260b.

MiRNAs have been reported to modulate multiple biological process pathways. In this study, we performed a functional enrichment analysis of DEMs using GO terms and KEGG. Consequently, these miRNAs were predicted to be involved in numerous neurological diseases-related pathways. Further research on exploring these miRNAs target genes is worthwhile.

Even though our work provides evidence to support the potential role of miRNAs as a suitable diagnosis and prognosis biomarker in patients with TDs, a larger TDs cohort is still needed for further verification.

## Figures and Tables

**Figure 1 brainsci-12-00829-f001:**
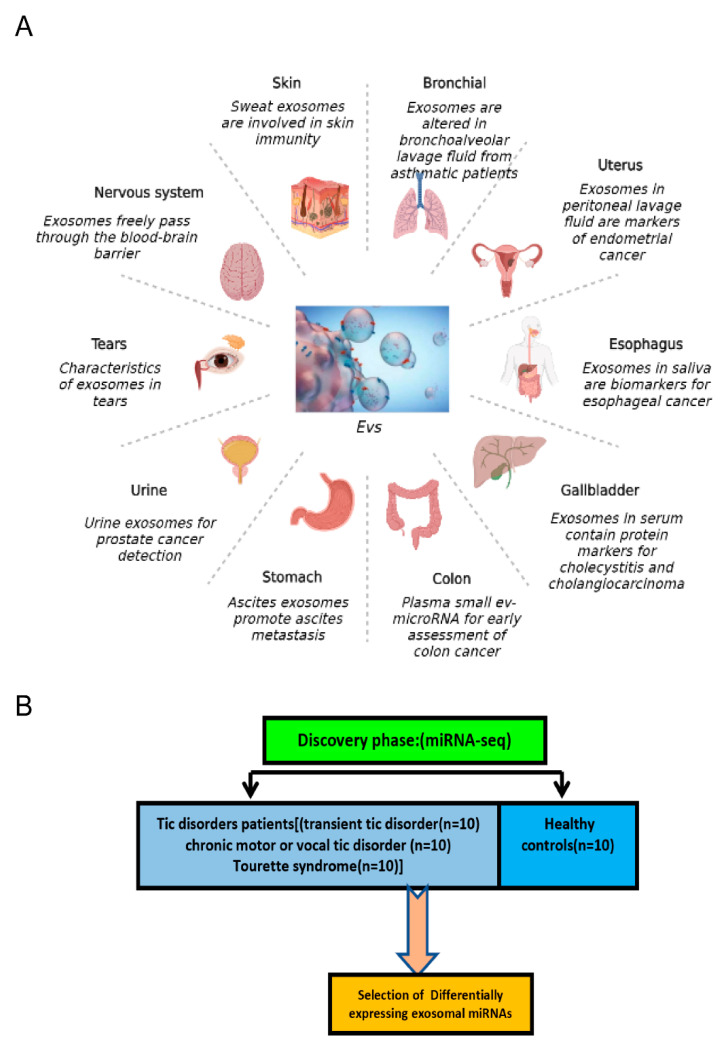
Schematic overview of study design. (**A**) Exosomes play an ir-replaceable role in various diseases in different systems. (**B**) A total of 10 children with transient tic disorders, 10 children with chronic motor or vocal tic disorder, 10 children with Tourette Syndrome, and 10 children who were neurological typical were included in the current research. RNA-seq was operated to identify small extracellular vesicle-derived miRNAs with differential expression levels and followed by bioinformatics analysis and functional prediction.

**Figure 2 brainsci-12-00829-f002:**
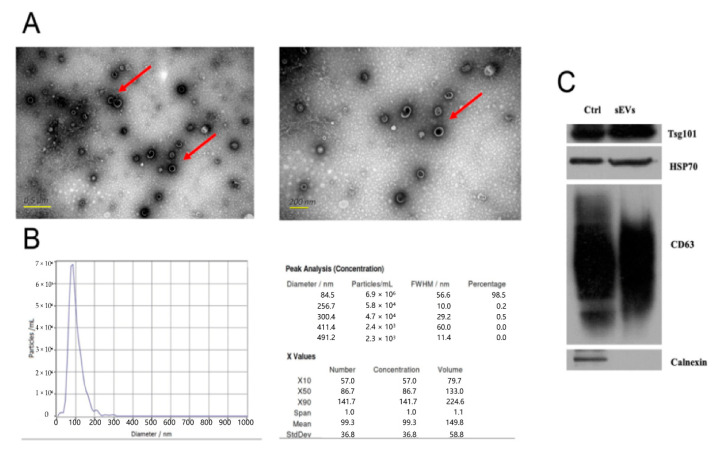
Exosomes derived from plasma supernatant were identified. (**A**) The morphology of sEVs was observed by transmission electron microscopy (TEM). The sEVs were highlighted by the red arrow. (**B**) sEVs concentrations and sizes were measured by nanoparticle tracking analysis (NTA) and the value was presented as the mean ± SD in the right table. (**C**) The positive biomarkers (CD63, Tsg101, and HSP70) and negative biomarker (Calnexin) of sEVs were detected by Western blot analysis, and the cell lysates of mesenchymal stem cell (MSC) were extracted as control.

**Figure 3 brainsci-12-00829-f003:**
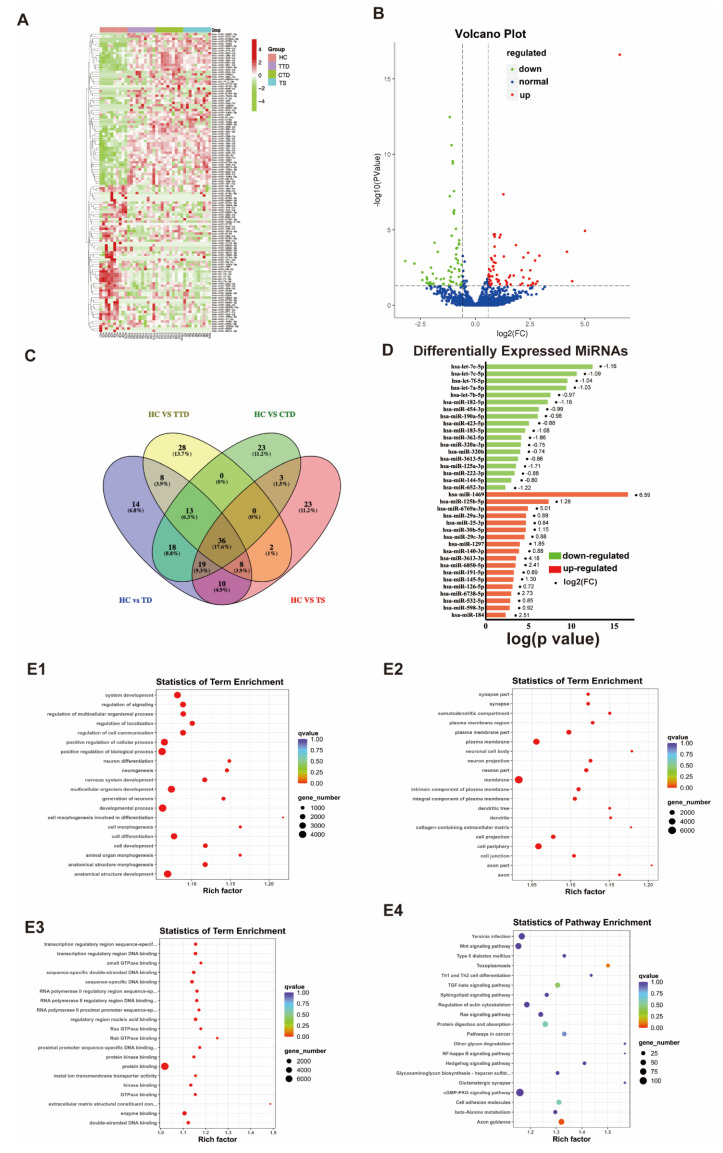
Expression Profile of miRNAs derived from plasma sEVs in tic disorder patients. (**A**) Heat map representing color-coded expression levels of differentially expressed miRNAs (DEMs), green: downregulated, red: upregulated. The scale of the color-bar on the right side is based on the z-score. Micro-RNAs’ names are given along the same side. (**B**) Volcano plot analysis. The red dots represent upregulated miRNAs, the green dots represent downregulated miRNAs. (**C**) The Venn-diagram illustrates the numbers of differentially expressed miRNAs in the four comparisons, including healthy controls (HCs) versus tic disorder children (TD), HCs versus transient tic disorder patients (TTD), HCs versus chronic motor or vocal tic disorder patients (CTD), and HCs versus Tourette Syndrome patients (TS). The coincident part indicates 36 DEMs shared between the four comparisons. (**D**) Bar chart analysis of differentially expressed miRNAs. The red color represents upregulated miRNAs, the green color represents downregulated miRNAs, the point represents the value of the log2 (FC). (**E**) The bubble plot. (**E1**−**E3**) shows GO terms distribution of all 126 DEMs-target mRNAs in these four comparisons; bubble color and size correspond to the Q value and DEMs number enriched in each term. (**E4**) represents KEGG pathway analysis of 126 DEMs.

**Figure 4 brainsci-12-00829-f004:**
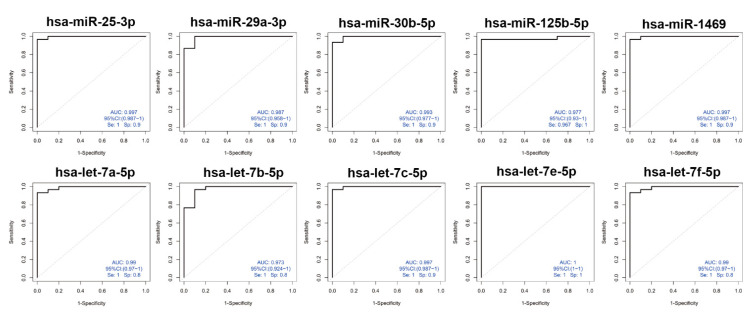
Receiver operating characteristic (ROC) analyses of differentially expressed miRNAs in TDs and controls. The 10 ROC curves show the ROC and AUC scores for the top ten dysregulated miRNAs. Upper panel: upregulated has-miR-25-3p, has-miR-29a-3p, has-miR-30-5p, has-miR-125b-5p, has-miR-1469. Lower panel: downregulated has-let-7a-5p, has-let-7b-5p, has-let-7c-5p, has-let-7e-5p, has-let-7f-5p.

**Figure 5 brainsci-12-00829-f005:**
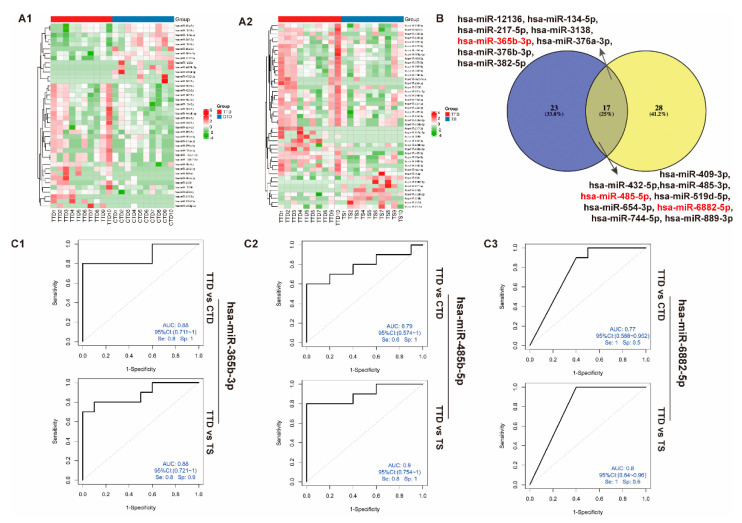
Identification and characterization of differentially expressed miRNAs in TTD and matched CTD or TS. (**A**) The heat map analysis identifies the significant differentially expressed miRNAs (DEMs). The two heat maps, (**A1**) and (**A2**), represent transient tic disorder patients (TTD) versus chronic motor or vocal tic disorder patients (CTD) and TTD versus Tourette Syndrome patients (TS), respectively. The color bar on the right side represents the scale for the Z-score. The red color indicates high expression, and the green color indicates low expression. Representative micro-RNAs are shown on the right side of the heat map. (**B**) The Venn-diagram for the two comparisons TTD versus CTD and TTD versus TS. The coincident part indicates 17 DEMs shared between the two comparisons. (**C**). Receiver operating characteristic (ROC) analyses of differentially expressed miRNAs in TTD, CTD, and TS. Three miRNAs, including hsa-miR-365b-3p (**C1**), hsa-miR-485b-5p (**C2**), and hsa-miR-6882-5p, were selected from the coincident part DEMs between the two comparisons (**C3**).

**Table 1 brainsci-12-00829-t001:** Clinical features of all participants.

Discovery Set				
	TD			HCs
	TS	TTD	CTD	
NO	10	10	10	10
Female: Male	2:8	3:7	0:10	3:7
Age, Mean ± SD	9.01 ± 2.84	6.76 ± 1.61	9.42 ± 1.78	9.52 ± 2.86
Age at first attack, Mean ± SD	6.06 ± 1.57	6.50 ± 1.41	7.71 ± 1.98	NA

Age and age at first attack used were presented as mean ± standard deviation. Abbreviation: tic disorder (TD), Tourette Syndrome (TS), transient tic disorder (TTD), chronic motor or vocal tic disorder (CTD), Healthy controls (HCs), NA (Not Available).

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
