# Peer review of "Micro-RNAs from Plasma-Derived Small Extracellular Vesicles as Potential Biomarkers for Tic Disorders Diagnosis"

_brainsci, 2022, doi:10.3390/brainsci12070829_

Round 1

Reviewer 1 Report

The manuscript of Yilong Wang et al. describes the characterization of specific miRNAs derived from plasma small extracellular vesicles of patients with tic disorders versus healthy controls and their potential use as biomarkers for diagnosis and prognosis of tic disorders. Although the studied cohorts are small-sized as stated in the discussion as one of the study's limitations, the study is innovative, the methods used are sound and clearly specified, the results quite original, the discussion concise and the references are the most recent. The overall manuscript contains important original research findings and it is concise and  well written, therefore it deserves publication if certain points are addressed.

Points that need revision:

1. Line 86 (page 2): More general information about the role of EVs and their typical cargo content (including miRNAs), such as lncRNAs, DNA, peptides, lipids, etc, is required. Also, as the article mentions several of them in the results and discussion, more information concerning EV surface proteins would be beneficial. After providing this information, the authors may state that while miRNA protection is not their primary function, EVs may be protective of their enclosed miRNA content.

2. Some typos need correction. For example

-Line 95 (page 2): typo - healthy controls* not control

-Line 293: (page 8): typo - We then analysed* not analysis

3. Materials and methods: 2.2. Which EV sizes are excluded according to the study’s criteria?

4. 2.4-2.9: a little more information on the purpose of each method would be helpful for readers and fellow scientists who would be interested in following a similar protocol.

5. Figure 3.: Some of the figure’s details are not quite readable. Also the heat map is not readable at all.

6. Lines 370-372 (page 10): vague meaning - needs to be rephrased.

7. Figure 5: In the last sentence of the legend, probably wrong miRNAs are mentioned.

Author Response

Dear Editor,

  We would like to thank the anonymous reviewers for their kind comments and valuable suggestions. We carefully read the reviewers’ reports and revised the manuscript according to their suggestion. All changes to the original manuscript in red front.

  Following are comments from reviewer #1 and our responses to the comments:

Reviewer #1:

  1. Line 86 (page 2): More general information about the role of EVs and their typical cargo content (including miRNAs), such as lncRNAs, DNA, peptides, lipids, etc, is required. Also, as the article mentions several of them in the results and discussion, more information concerning EV surface proteins would be beneficial. After providing this information, the authors may state that while miRNA protection is not their primary function, EVs may be protective of their enclosed miRNA content.

Answer:Thank the reviewer for his kind suggestion. The information of EVs has been improved and shown in red font in third paragraph (line 90-95, page 2) of Introduction part.

  1. Some typos need correction. For example

-Line 95 (page 2): typo - healthy controls* not control

-Line 293: (page 8): typo - We then analysed* not analysis

Answer:Thank the reviewer for his kind suggestion. The wrong typos have been corrected and shown in red.

  1. Materials and methods: 2.2. Which EV sizes are excluded according to the study’s criteria?

Answer:Thank the reviewer for his kind suggestion. In results of NTA, the particle-size distribution of EVs obtained in this study has a reasonable range from 35-185nm and its major peak is around 100nm. Thus, we didn’t offer additional exclusion of EVs. 

  1. 2.4-2.9: a little more information on the purpose of each method would be helpful for readers and fellow scientists who would be interested in following a similar protocol.

Answer:Thank the reviewer for his kind suggestion. More information has been added in corresponding chapters and shown in red.

  1. Figure 3.: Some of the figure’s details are not quite readable. Also the heat map is not readable at all.

Answer:Thank the reviewer for his kind suggestion. The Figure 3 has been replaced by a more clear one.

  1. Lines 370-372 (page 10): vague meaning - needs to be rephrased.

Answer:Thank the reviewer for his kind suggestion. The last paragraph in Discussion part (Lines 394-396 ,Page 11 ) has been rephrased for better understanding and shown in red.

  1. Figure 5: In the last sentence of the legend, probably wrong miRNAs are mentioned.

Answer:Thank the reviewer for his kind suggestion. Wrong miRNAs information has been corrected and shown in red.

Reviewer 2 Report

Summary: This study aims to identify the miRNA-based biomarkers for tic disorders. The group tested the plasma-derived small extracellular vesicles instead of conventional plasma. I opine that this study is highly relevant and timely, given there is a lack of molecular diagnostics for TDs yet. Overall, the study design is elegant and robust. The manuscript is well written, and I have a few minor comments below.

Comments and Suggestions:

  • Specify the sequencing depth (x million reads per sample) used in the study.
  • There were more male participants than female participants in this study. Please add a discussion about the potential sex differences.
  • Figure 1 is not necessary/informative in the present form. If authors could add more graphical features like cartoons of miRNAs or EVs, that would be more helpful.
  • In figure 2A, the scale bar is barely visible. Please change the color to yellow.
  • The fold change threshold (>1.5) used for identifying differentially expressed miRNAs is considered weak. Please repeat the whole analysis related to figure 3 and 5 using the fold change > 2. If not, please provide the rationale for using the specified cutoff.
  • Please show the fold change values in Figure 3D in addition to the p-Value
  • Add discussion about why miRNAs from plasma-derived small extracellular vesicles are superior over just plasma.

Author Response

Dear Editor,

  We would like to thank the anonymous reviewers for their kind comments and valuable suggestions. We carefully read the reviewers’ reports and revised the manuscript according to their suggestion. All changes to the original manuscript in red front.

  Following are comments from reviewer #2 and our responses to the comments:

Reviewer #2

  • Specify the sequencing depth (x million reads per sample) used in the study.

Answer:Thank the reviewer for his kind suggestion. Approximately 20 million reads were generated per sample in our study. This information has been added to the Results part 3.3 and shown in red.

  • There were more male participants than female participants in this study. Please add a discussion about the potential sex differences.

Answer:Thank the reviewer for his kind suggestion. The discussion about the potential sex differences has been added in the Results part 3.1 and shown in red.

  • Figure 1 is not necessary/informative in the present form. If authors could add more graphical features like cartoons of miRNAs or EVs, that would be more helpful.

Answer:Thank the reviewer for his kind suggestion. Figure 1 has been re-designed and the old one has been replaced.

  • In figure 2A, the scale bar is barely visible. Please change the color to yellow.

Answer:Thank the reviewer for his kind suggestion. The color of scale bar has been changed to yellow.

  • The fold change threshold (>1.5) used for identifying differentially expressed miRNAs is considered weak. Please repeat the whole analysis related to figure 3 and 5 using the fold change > 2. If not, please provide the rationale for using the specified cutoff.

Answer:Thank the reviewer for his kind suggestion. sEVs derived MiRNAs abundance is relatively low in human plasm samples, thus, a 1.5-fold change (FC) in miRNAs expression level could be a cutoff for establishing differential change. The rationale and related references have been added in the Result part 3.3(line 257-259, page 7) and shown in red.

  • Please show the fold change values in Figure 3D in addition to the p-Value

Answer:Thank the reviewer for his kind suggestion. The Log2(FC) has been added in Figure 3D.

  • Add discussion about why miRNAs from plasma-derived small extracellular vesicles are superior over just plasma.

Answer:Thank the reviewer for his kind suggestion. The discussion about the advantages has been added in the Discussion part and show in red.

Reviewer 3 Report

This article describes a study assessing and comparing miRNA profiles in TTD, CTD, TS, and HC groups. Although an area of interest in the field, the paper would improve by further clarification of the selection of subjects and a more in depth discussion on how these findings compare to previous miRNA studies (such as Rizzo et al 2015) as well as more discussion on the limitations of this study. The design of comparing TTD to other TD groups is limited as we do not know if these TTD may go on to meet CTD criteria and thus I have concerns that the goal of comparing TTD to other TD was not accomplished.

Specifically, I have the following comments/suggestions:

Intro:

-claims that early intervention and therapy guarantee superior QoL have no reference to support them

-statement that it is difficult to diagnose TDs should be supported with a reference 

Methods:

-how were HC recruited? Were they screened for ADHD or OCD?

-co-occurring neuropsychiatric symptoms are discussed throughout the paper but the presence of these diagnoses or dimensional ratings of symptom severities are not presented on the subjects. if available, please include. if not, this should be noted as a limitation 

Results:

-Figure 3 seems to show the same data in multiple ways. would recommend choosing one method to show

-As noted above, it is not clear that the TTD to other CTD comparison is valid as it is not clear that the TTD group will not go on to be a CTD individual. Would be more valid if it was a child with tics <1 year with the miRNA assessment a few years after last tics seen. 

Discussion

-should include how your findings compare to other miRNA in TS studies

Author Response

Dear Editor,

  We would like to thank the anonymous reviewers for their kind comments and valuable suggestions. We carefully read the reviewers’ reports and revised the manuscript according to their suggestion. All changes to the original manuscript in red front.

  Following are comments from reviewer #3 and our responses to the comments:

Reviewer #3

Specifically, I have the following comments/suggestions:

Intro:

-claims that early intervention and therapy guarantee superior QoL have no reference to support them

Answer:Thank the reviewer for his kind suggestion. The references have been added in first paragraph in the Introduction part (line 54-59, page 2) and shown in red.

-statement that it is difficult to diagnose TDs should be supported with a reference 

 Answer:Thank the reviewer for his kind suggestion. This statement has been rephrased and references have been added in the Introduction part (line 61-65, page 2) and shown in red.

Methods:

-how were HC recruited? Were they screened for ADHD or OCD?

Answer:Thank the reviewer for his kind suggestion. The healthy control subjects were recruited from the Children’s Hospital of Zhejiang University School of Medicine, and confirmed healthy and neurologically normal by medical history, general examinations, laboratory examinations, and all had been excluded from ADHD and OCD screening.

-co-occurring neuropsychiatric symptoms are discussed throughout the paper but the presence of these diagnoses or dimensional ratings of symptom severities are not presented on the subjects. if available, please include. if not, this should be noted as a limitation 

Answer:Thank the reviewer for his kind suggestion. Our study does not involve the diagnoses or dimensional ratings of symptom severities. The introduction of the co-occurring neuropsychiatric symptoms is supposed to highlight the complexity of TDs’ clinical manifestations and diagnosis. According to the suggestion of the reviewer, the description about co-occurring neuropsychiatric symptoms has been deleted in the Discussion part.

Results:

-Figure 3 seems to show the same data in multiple ways. would recommend choosing one method to show

Answer:Thank the reviewer for his kind suggestion. The data in Figure 3 is not same. It described the distribution of differentially expressed miRNAs (DEMs) in different comparisons and demonstrated different information about DEMs. It may be difficult for us to delete any part of it.

-As noted above, it is not clear that the TTD to other CTD comparison is valid as it is not clear that the TTD group will not go on to be a CTD individual. Would be more valid if it was a child with tics <1 year with the miRNA assessment a few years after last tics seen. 

Answer:Thank the reviewer for his kind suggestion. Most of the TTD children selected in this study had their first onset time in 2020 and the first half of 2021. After the samples were collected, we followed up by phone calls before the experiment started around March this year, and confirmed that none of the children who had been enrolled had a recurrence. And this week, we followed up again, and none of the children enrolled in the TTD group had relapsed.

Discussion

-should include how your findings compare to other miRNA in TS studies

Answer:Thank the reviewer for his kind suggestion. The comparison between our fingdings and other miRNA in TS studies has been added in the forth paragraph of the discussion part (line 381-388, page 11) and shown in red.
